# Formate Utilization by the Crenarchaeon *Desulfurococcus amylolyticus*

**DOI:** 10.3390/microorganisms8030454

**Published:** 2020-03-23

**Authors:** Ipek Ergal, Barbara Reischl, Benedikt Hasibar, Lokeshwaran Manoharan, Aaron Zipperle, Günther Bochmann, Werner Fuchs, Simon K.-M. R. Rittmann

**Affiliations:** 1Archaea Physiology & Biotechnology Group, Department of Functional and Evolutionary Ecology, Universität Wien, 1090 Wien, Austria; ipek.ergal@univie.ac.at (I.E.); barbara.reischl@univie.ac.at (B.R.); Lokeshwaran.Manoharan@nbis.se (L.M.); aaronzipperle@gmail.com (A.Z.); 2Department IFA Tulln, Institute for Environmental Biotechnology, University of Natural Resources and Life Sciences, 1180 Wien, Austria; benedikt.kleibel@boku.ac.at (B.H.); guenther.bochmann@boku.ac.at (G.B.); werner.fuchs@boku.ac.at (W.F.)

**Keywords:** Archaea, Crenarchaeota, anaerobe, microbial physiology, metabolism, one-carbon, formaldehyde

## Abstract

Formate is one of the key compounds of the microbial carbon and/or energy metabolism. It owes a significant contribution to various anaerobic syntrophic associations, and may become one of the energy storage compounds of modern energy biotechnology. Microbial growth on formate was demonstrated for different bacteria and archaea, but not yet for species of the archaeal phylum Crenarchaeota. Here, we show that *Desulfurococcus amylolyticus* DSM 16532, an anaerobic and hyperthermophilic Crenarchaeon, metabolises formate without the production of molecular hydrogen. Growth, substrate uptake, and production kinetics on formate, glucose, and glucose/formate mixtures exhibited similar specific growth rates and similar final cell densities. A whole cell conversion experiment on formate revealed that *D. amylolyticus* converts formate into carbon dioxide, acetate, citrate, and ethanol. Using bioinformatic analysis, we examined whether one of the currently known and postulated formate utilisation pathways could be operative in *D. amylolyticus*. This analysis indicated the possibility that *D. amylolyticus* uses formaldehyde producing enzymes for the assimilation of formate. Therefore, we propose that formate might be assimilated into biomass through formaldehyde dehydrogenase and the oxidative pentose phosphate pathway. These findings shed new light on the metabolic versatility of the archaeal phylum Crenarchaeota.

## 1. Introduction

Formic acid, a monocarboxylic acid, is one of the simplest organic compounds. It is a colourless liquid with a pungent odour [1]. The freezing and the boiling points of formate are 8.3 and 100.7 °C, respectively. The carbon of formic acid is a poor electrophile. This characteristic makes formic acid a strong acid (pK_a_ = 3.75), e.g., compared to acetic acid (pK_a_ = 4.75) [2]. The salt of formic acid is formate. Formate can be produced using various routes such as the electrochemical reduction of CO_2_ [3,4,5,6,7], photo-reduction of CO_2_ [8], hydrogenation of CO_2_ [9,10], selective oxidation of biomass [11,12], partial oxidation of natural gas [13], and hydration of syngas (e.g., carbon monoxide) [14]. In 1670, John Wray had already isolated formate from ants [15], and later it was named after “formicidae”, the ant family. Formate has numerous functions in biological systems, such as serving as an irritant in the sprayed venom of some ant species [16], as antibacterial substance [17], and it is used for food preservation, as cosmetic additive, or as pesticide [18]. Moreover, it can be used as a non-flammable liquid fuel [19], an energy storage compound [20,21], and as a feedstock for cultivation of microorganisms [22]. Hence, biological production and utilization of formate is of ecological and biotechnological significance.

Metabolisation of formate can occur during acetogenesis [23], methanogenesis [24], and molecular hydrogen (H_2_) production [25]. Moreover, formate can provide cells with both a carbon source and reducing power. It delivers a significant contribution to anaerobic syntrophic associations, as formate is transferred as redox currency between the organisms [26]. According to a search using the PubChem Substance and Compound Databases [27], there are 189 biological systems that include formate as a reaction component. This shows the significance of formate in functioning of biological systems. Therefore, formate assimilation pathways and studying how microorganisms with different physiologies can convert formate into specific metabolites are increasingly attracting attention.

One possible way to biologically produce formate is the reduction of CO_2_ by H_2_ through the hydrogen-dependent CO_2_ reductase (HDCR). The HDCR was first isolated and characterized in *Acetobacterium woodii* [28]. It consists of four subunits: a putative formate dehydrogenase (FdhF1/2), a Fe–Fe hydrogenase (HydA), and two small electron transfer subunits [28,29]. The HDCR operates the direction of the catalysis depends on the substrate concentration [28].

Another option to biologically produce formate is through pyruvate cleavage by pyruvate formate lyase (PFL) (30). PFL is an oxygen sensitive, ubiquitous enzyme that supports increased ATP yield during fermentation of, e.g., glucose [14]. PFL plays a central role in many organisms providing formate and acetyl-coenzyme A (CoA) via the conversion of pyruvate and CoA. The activation of PFL occurs with an enzyme referred to as PFL-activating enzyme (PFL-AE). The activation of PFL involves an abstraction of hydrogen through PFL-AE, which belongs to the radical S-adenosyl-methionine (SAM) super-family [30,31]. To understand the assimilation of formate and how it can support microbial growth, the reverse PFL reaction was studied in vivo in *Escherichia coli.* [32]. PFL-dependent co-assimilation of acetate and formate was demonstrated with *E. coli* mutant strains. It was concluded that PFL could support growth on formate as the carbon source.

Oxidation of formate can be catalysed by some microorganisms such as Burkholderiaceae [33], Clostridiaceae [34,35], and Enterobacteriaceae [36,37,38,39,40,41,42,43,44]. The oxidation of formate is performed to avoid cellular toxification through intracellular formate accumulation [45]. In *E. coli*, formate is able to disturb the membrane potential and allows the hydrogen ion (H^+^) to enter the cell from the medium [45,46]. At certain concentrations, this leads to a collapse of the pH gradient across the cytoplasmic membrane, referred to as uncoupling, that consequently ceases the metabolism [47]. Hence, detoxification mechanisms were developed to avoid uncoupling at high concentrations of formate. In *E. coli*, formate is disproportionated into H_2_/CO_2_ by the formate hydrogen lyase (FHL) complex. There, Fdh along with hydrogenase-3 (Hyd-3) which is encoded by *hyc* operon, can form the FHL complex [48]. 

Even though the oxidation of formate can occur as a result of detoxification mechanism, formate cannot be metabolised by members of Burkholderiaceae, Clostridiaceae, and Enterobacteriaceae, as this reaction is not energetically sufficient to support growth. Under anoxic conditions, the stoichiometry of formate conversion is given below (Equation (1)):HCOO^−^ + H_2_O → HCO_3_^−^ + H_2_ ΔG0′ = +1.3 kJ mol^−1^(1)

Until the discovery of the hyperthermophilic archaeon *Thermococcus onnurineus* NA1 [49,50], it was assumed that the reaction is only thermodynamically possible when a methanogenic or sulphate-reducing partner is present to remove H_2_, which is one of the end products of syntrophic formate oxidation [51,52,53,54]. *T. onnurineus* is able to oxidise formate into H_2_/CO_2_, and coupling this reaction to chemiosmotic ATP synthesis [25,55]. The metabolic pathway responsible for formate oxidation in *T. onnurineus* was characterised by using proteomics [56]. It was found that formate oxidation proceeds via a membrane-bound enzyme system, comprised of Fdh, and a membrane-bound hydrogenase (Mfh), a sodium-proton (Na^+^/H^+^) antiporter (Mnh) and a Na^+^-dependent ATP synthase [57]. Furthermore, it was revealed that several copies of hydrogenase gene clusters (*fdh-mfh-mnh*) are present in *T. onnurineus.* The hydrogenase genes in *fdh2-mfh2-mnh2* were upregulated more than 2-fold when formate was provided as sole energy source to *T. onnurineus*, which is an indication of the importance of the hydrogenase genes in the *fdh2-mfh2-mnh2* gene cluster for growth coupled to formate oxidation and H_2_ production [25]. 

Recently, a novel synthetic formate-fixing pathway was proposed, which is involved in the acetyl-CoA exchange to formyl-CoA and formate reduction to formaldehyde [58]. Formaldehyde can be integrated into the central carbon metabolism much easier than direct formate utilization, since it is a highly reactive compound [59]. However, the reduction potential of formate to formaldehyde is quite low under standard biochemical conditions [−650 mV≤ E°′≤−450 mV (6 ≤ pH ≤ 8; 0 ≤ I ≤ 0.25 mol L^−1^)] [60]. Hence, the activation of formate requires an electron donor such as universal electron carrier NAD(P)H (−370 mV ≤ E′ ≤ −280 mV) [58]. One of the formate-fixing reactions is catalysed through the formaldehyde dehydrogenase (FoDH) enzyme, which directly converts formate to formaldehyde using NADH as a cofactor [61,62]. Investigating formate assimilation in other organisms could contribute to the identification of new formate-fixation pathways and its enzymes.

*Desulfurococcus amylolyticus* DSM 16532 [63] is an anaerobic, hyperthermophilic Crenarchaeon able to grow on a broad range of polymers and sugars [63,64]. Recently, physiological variables, such as the specific growth rate (µ), were determined for of *D. amylolyticus* grown on fructose or glucose in chemically defined medium [65]. A metabolic reconstruction revealed that *D. amylolyticus* contains all glucose metabolism-related genes and harbours several genes for H_2_ production, such as pyruvate ferredoxin oxidoreductase, glyceraldehyde-3-phosphate ferredoxin oxidoreductase, and two membrane-bound hydrogenases [64,65]. As growth on formate coupled to H_2_ production by hyperthermophilic Archaea could be an opportunity in biotechnology [66,67], we investigated whether *D. amylolyticus* could be used for this purpose.

In this work, we examined the substrate uptake, growth, and production kinetics of *D. amylolyticus* grown on formate, glucose, and a mixture of glucose/formate in closed batch cultivation mode. The intention was to physiologically characterise the organism with respect to μ, cell-specific H_2_ and CO_2_ productivities, and maximum cell concentration. After this, the mass balance analysis of substrate uptake and product formation kinetics was performed in whole cell conversion experiments. Finally, we investigated and compared the metabolic capacity for formate utilization of *D. amylolyticus* to other formate metabolising microorganisms on the genomic level with the goal of revealing possible formate assimilation pathways.

## 2. Material and Methods

### 2.1. Chemicals

The standard test gas utilized in gas chromatography (GC) comprised the following composition: 0.01 Vol.-% CH_4_; 0.08 Vol.-% CO_2_ in N_2_ (Messer GmbH, Wien, Austria). All other chemicals were of highest grade available. H_2_, CO_2_, N_2_, 20 Vol.-% CO_2_ in N_2_, and CO were of test gas quality (Air Liquide, Schwechat, Austria).

### 2.2. Microorganism and Medium Composition

*Desulfurococcus amylolyticus* DSM 16532 [63,68] was purchased from the Deutsche Sammlung von Mikroorganismen und Zellkulturen GmbH (DSMZ). The medium was prepared as previously described [64,65]. A modified DSMZ medium, No. 395, without yeast extract, powdered sulphur and glucose was prepared when only formic acid containing medium was used for growth of *D. amylolyticus*. The medium contained (per L): 0.33 g of NH_4_Cl; 0.33 g of KH_2_PO_4_; 0.33 g of KCl; 0.44 g of CaCl_2_·2H_2_O; 0.70 g of MgCl_2_·6H_2_O; 0.50 g of NaCl; 0.80 g of NaHCO_3_; 0.50 g of Na_2_S·9H_2_O; 1 mL of trace elements SL-10; and 10 mL of vitamin solution as previously described [64,65].

### 2.3. Closed Batch Cultivations

Closed batch cultivations were conducted in two different sets of experiments. The first set of experiments were designed as “end-point experiments” where growth was monitored at each time-point, whereas substrate uptake and production measurements were only performed at end of the experiment. The second set of experiments were designed as “time-point experiments” where substrate uptake, growth and production kinetics were observed and measured at each time-point during the experiment.

Cultures of *D. amylolyticus* were grown anaerobically at 0.2–0.3·10^5^ Pa in a 100 Vol.-% N_2_ atmosphere in a closed batch set-up as previously described [64,65]. Concentrations of carbon sources were adjusted to the same carbon concentration (C-mmol L^−1^). For end-point experiments the following carbon sources were individually tested: formic acid and glucose at 116.6 C-mmol L^−1^, respectively. Formic acid at a concentration of 50 C-mmol L^−1^, and glucose at a concentration of 66.6 C-mmol L^−1^ (total 116.6 C-mmol L^−1^) were added for the co-substrate experiment. Additionally, glucose and formic acid were tested as carbon sources at 66.6 and 50 C-mmol L^−1^, respectively, to assess the difference in substrate uptake, growth and production kinetics between the co-substrate and single substrate experiment. 

The pre-culture for inoculation was obtained from a formic acid pre-grown *D. amylolyticus* culture. All experiments were performed in quadruplicates together with a negative (un-inoculated) control and reproduced twice. Moreover, we performed an additional positive–negative (inoculated on medium without formic acid) control to observe growth kinetics on medium containing just vitamins and trace elements, without formic acid (Appendix A). Pressure measurements of the serum bottle headspace were performed with a digital manometer (LEO1-Ei, −1-3bar rel, Keller, Germany) and the measurements were performed before samples were taken for microscope analysis. For time-point experiments, formic acid was used as the carbon source at concentration of 100 C-mmol L^−1^. For each time-point, identical sets of serum bottles (quadruplicates) were prepared and inoculated. They were not manipulated until the destructive gas chromatography (GC) measurement (see below in GC section).

### 2.4. Whole Cell Conversion Experiments

*D. amylolyticus* was pre-grown on formic acid and harvested by centrifugation (Eppendorf Centrifuge 5415R, Eppendorf, Hamburg, Germany) for 20 min and 15,700 g. The supernatant was removed and the resulting pellet was washed with the respective medium. After the washing step, the cells were resuspended in buffer containing (per L): NH_4_Cl 0.33 g; KH_2_PO_4_ 0.33 g; KCl 0.33 g; CaCl_2_·2H_2_O 0.44 g; MgCl_2_·6 H_2_O 0.70 g; NaCl 0.50 g; NaHCO_3_ 0.80 g. Serum bottles of 120 mL were supplemented with formic acid (concentration of 20, 50, and 100 C-mmol L^−1^). Cultures (1·10^8^ cells mL^−1^) were incubated for 5 and 12 h, respectively and GC analyses were immediately conducted afterwards. All experiments were performed in triplicates together with a negative (un-inoculated) control.

### 2.5. Cell Counting

*D. amylolyticus* cells were counted using a Nikon Eclipse 50i microscope (Nikon, Amsterdam, Netherlands) at each sampling point. The samples for cell count were taken from each individual closed batch run using syringes (Soft-Ject, Henke Sass Wolf, Tuttlingen, Germany) and hypodermic needles (Sterican size 14, B. Braun, Melsungen, Germany). An amount of 10 µL of sample was applied onto a Neubauer improved cell counting chamber (Superior Marienfeld, Lauda-Königshofen, Germany) with a grid depth of 0.1 mm.

### 2.6. Gas Chromatography

Time-point and end-point GC measurements were performed from serum bottles that remained without any manipulation after inoculation, except incubation in air bath until the GC measurement was performed. To analyse the gas composition inside the serum bottles from end-point and time-point experiments, destructive sampling was employed. The gas compositions were analysed using a GC (7890A GC System, Agilent Technologies, Santa Clara, CA, USA) with a 19808 Shin Carbon ST Micropacked Column (Restek GmbH, Bad Homburg, Germany) and provided with a gas injection and control unit (Joint Analytical System GmbH, Moers, Germany) as described earlier [64].

### 2.7. Formic Acid Analysis

The Formic Acid Assay Kit (Megazyme Inc., Bray, Ireland) was used for measurements of formic acid concentrations in samples which were previously diluted to the linear range of the assay kit to yield a formic acid concentration of 0.004–0.200 g L^−1^. The microplate (Crystal Clear, Greiner Bio One) assay was applied according to manufacturer’s instructions: a 255 μL reaction volume.

### 2.8. HPLC

The determination of sugars, volatile fatty acids (VFAs), organic acids, and alcohols were performed with high performance liquid chromatography (HPLC) system (Agilent 1100), consisting of a G1310A isocratic pump, a G1313A ALS autosampler, a Transgenomic ICSep ICE-ION-300 column, a G1316A column thermostat set at 45 °C, a G1362A RID refractive index detector, measuring at 45 °C (all modules were from Agilent 1100 (Agilent Technologies, CA, USA)). The measurement was performed with 0.005 mol L^−1^ H_2_SO_4_ as solvent, with a flow rate of 0.325 mL min^−1^ and a pressure of 48–49 bar. The injection volume was 40 µL.

### 2.9. Data Analysis

For the quantitative analysis, the maximum specific growth rate (µ_max_ [h^−1^]) and mean specific growth rate (µ_mean_ [h^−1^]) were calculated as follows: N = N^0^·e^µt^ with N, cell number [cells mL^−1^]; N^0^, initial cell number [cells mL^−1^]; t, time [h] and e, Euler number. According to the delta cell counts in between sample points, µ was assessed. The CO_2_ evolution rate (CER [mmol L^−1^ h^−1^] (C-molar)), the cell specific CO_2_ productivity (qCO_2_, cell [mmol cell^−1^ h^−1^] (C-molar)) [67] was calculated from the end point gas composition of the non-manipulated serum bottles. The ash content and elementary composition of *D. amylolyticus* were presumed to relate to published results [69]. The elementary composition was used for the calculation of the mean molar weight, Carbon balance (C-balance) and the degree of reduction (DoR) balance of the corresponding biomass. DoR denotes the number of electrons an atom can donate, summed up for all the atoms of a molecule, or biomass elementary composition, divided by the number of carbon atoms in the molecule/biomass elementary composition. Yields of by-products were determined after HPLC measurement. The values were normalized according to zero (time-point zero) control and negative values (if there were any) were assumed as zero.

### 2.10. Gibbs Free Energy Calculations

Standard Gibbs free energy change (ΔG^0′^) is used to describe whether a certain chemical reaction can be utilized for microbial energy conservation. However, the underlying thermodynamic calculations are usually standardized to 25 °C and 1 bar pressure. For thermophilic bioprocesses the physiological conditions differ significantly and, consequently, values have to be adapted, as especially temperature has a huge impact on thermodynamics [70]. In the present study, *D. amylolyticus* was cultivated at 80 °C. The recalculation method applied in the current study is based on previously published results [70], which provide standard state thermodynamic properties at temperatures up to 200 °C for a wide variety of anaerobic metabolic reactions. Moreover, they discuss the thermodynamic framework in detail and the application of the revised Helgeson Kirkham Flowers (HKF) equation of state to obtain standard molal Gibbs free energies of formation (G_f_^0^) at elevated temperatures and pressures for individual aqueous compounds. With one important exemption, the values for G_f_^0^ are taken from this paper. Unfortunately, the named study does not include data for formaldehyde, which were therefore obtained from another publication [71], and recalculated as specified. Finally, the Gibbs values for the overall reaction at standard concentrations (1 mol L^−1^) and pH of 7 (∆G^0′^) were calculated as previously described [70].

### 2.11. Genome Analysis

To investigate the formate metabolism of *D. amylolyticus*, the protein sequences from the whole genome of organisms where formate related pathways were previously described like *T. onnurineus* (Ton_GCF_000018365.1_ASM1836v1), *Pyrococcus furious* (Pfu_GCF_000007305.1_ASM730v1), *E. coli* K-12 (Eco_GCF_000750555.1_ASM75055v1), *Methanosarcina barkeri* (Mba_GCF_000195895.1_ASM19589v1), *Thermoanaerobacter kivui* (Tki_GCF_000763575.1_ASM76357v1) and *A. woodii* (Awo_GCF_000247605.1_ASM24760v1) were obtained from the NCBI RefSeq [72] database. Homologous proteins involved in formate-related metabolism of *D. amylolyticus* were identified by using Basic Local Alignment Search Tool (BLAST) [73] against the manually sorted proteins from characterised enzymes (Appendix A) with E-values and local identity cutoffs of <10^−10^ and >25%, respectively. Orthologous genes (genome level) were obtained by pair-wise all versus all BLAST of the aforementioned organisms using the "OrthoFinder" tool [74]. Furthermore, the orthologous gene (gene in a different species that evolved from a common ancestral gene by speciation) groups (ortho-groups) containing the characterised enzymes related to formate-related metabolism were retrieved. After sorting the proteins of interest, enzyme complexes were identified by using bidirectional BLAST. Results for query coverage, E-value, and identity can be found in Appendix A. In addition, the Pfam domains [75] of all the proteins in *D. amylolyticus* were also predicted for a further look into the domains of enzymes related to the formate metabolism.

## 3. Results

### 3.1. D. amylolyticus Grows on Formate

Recently, we investigated growth characteristics of *D. amylolyticus* on cellulose, fructose, arabinose, glucose, lactose, maltose, starch, and sucrose. Moreover, we partially re-annotated the genome and metabolically reconstructed the central carbon metabolism [65]. According to the metabolic reconstruction of *D. amylolyticus*, it seemed to be possible that the organism could grow through metabolisation of formate. To elucidate the growth kinetics of *D. amylolyticus* on formate and to be able to compare them to glucose and formate–glucose mixtures, end-point experiments were designed to analyse growth kinetics in defined medium with each of the substrates at the same concentration (166.6 C-mmol L^−1^). Additionally, formate and glucose were tested at concentrations of 50 and 66.6 C-mmol L^−1^, respectively, to be able to examine the growth kinetics at lower substrate concentrations. The results of the growth characteristics are shown in Figure 1. Cleary, *D. amylolyticus* did not grow on a medium where formic acid was omitted (Appendix A). The lag time lasted approximately 125 h, until growth of *D. amylolyticus* on each of the substrates commenced. The key physiological variables are presented in Table 1. The organism grew almost equally well on all substrates tested and at all concentrations. Astonishingly, the organism grew on formate as the sole energy source with a µ_max_ of 0.032 h^−1^. A slightly higher µ_max_ of 0.035 and 0.036 h^−1^ was only obtained on glucose and formate/glucose, respectively. The differences in µ_max_ might be explained by slightly different concentrations and hyperbolic relationship between µ and the substrate concentration [76,77]. In a previous study, we showed that *D. amylolyticus* comprises a µ_max_ of 0.059 h^−1^ at a concentration of 166.5 C-mmol L^−1^ glucose [65]. In our previous studies [64,65], growth of *D. amylolyticus* resulted in low cell densities only, a characteristic shared with many other hyperthermophilic microorganisms [78]. The growth of *D. amylolyticus* on glucose was accompanied by lactate, acetate, and formate production, whereas growth on formate resulted in production of acetate, citrate, and ethanol (Table 2).

Following this, we examined the gaseous product formation spectrum from all quadruplicate closed batch experiments on formate, glucose, and formate/glucose. CO_2_ was the only gas detectable in all growth experiments. The CO_2_ concentrations are shown in Figure 2, and an overview of CO_2_ productivities are presented in Table 2. In cultures which were grown on formate, the cumulative CO_2_ levels were in the ppm range. However, during growth on glucose or formate/glucose, the cumulative CO_2_ values were more than one order of magnitude higher. H_2_ could not be detected in any growth experiment, which is in contradiction to experiments in bioreactors [64], but in agreement to our previous closed batch experiments [65]. This indicates that *D. amylolyticus* did not to use the electrons from formate or glucose to balance homeostasis by producing H_2_. However, instead it could be possible that the electrons were used to balance anaplerotic reactions, which indicates that during formate metabolization, CO_2_ was assimilated into biomass through some of the several annotated CO_2_-fixing enzymes [65]. However, up to now it is not clear if the ATP for the CO_2_ fixation is retrieved from substrate level phosphorylation via glycolysis or from chemiosmotic ATP production or the pseudo-TCA cycle.

During the time-point experiments (Figure 3), the observed growth kinetics showed a similar trend as in the end-point experiments (Figure 1). Production and consumption of citrate and CO_2_ are shown in Figure 3 and an overview of CO_2_ productivities and mass balances are presented in Table 3. The results of the closed batch time-point experiment indicate that CO_2_-fixation occurs. Hence, during the first 300 h, CO_2_ was produced and subsequently consumed within the next 200 h of the experiment. We have previously shown that *D. amylolyticus* harbours several genes, which might be involved in CO_2_ fixation [65], and the results presented in this study suggest the possibility that CO_2_ might be fixed through enzymes of the reductive citric acid cycle, as citric acid is one of the produced metabolites and consumed compounds.

### 3.2. D. amylolyticus Converts Formate to CO_2_, Acetate, Citrate, and Ethanol

To investigate the metabolism on formate in more detail, whole cell conversion experiments were performed. A whole cell conversion experiment has unique advantages, such as minimising side reactions and avoiding biomass production [79,80,81]. Therefore, we performed experiments with high cell densities (1·10^8^ cells mL^−1^) using *D. amylolyticus* at formate concentrations of 20, 50, and 100 C-mmol L^−1^ in buffer. This experiment revealed astonishing findings. A summary of the obtained physiological variables is shown in Table 4. The highest formate uptake rate of 20.7 C-mmol L^−1^ was observed when *D. amylolyticus* was incubated at a concentration of 100 C-mmol L^−1^ formate. After 5 h of incubation, the production of small amounts of butyrate was detected in 50 and 100 C-mmol L^−1^ formate, and citrate, as well as ethanol, was only detected in one of the other. After 12 h of incubation, ethanol and low amounts of citrate were detected at all tested formate concentrations, but acetate was only detected at 100 C-mmol L^−1^ formate. The detection of citrate production during growing conditions and during the whole cell conversion experiment (compare Table 2 and Table 4) might indicate that the citric acid cycle was involved during formate assimilation. However, in our previous study, a canonical citric acid cycle or a reverse citric acid cycle could not be observed in the genome of *D. amylolyticus* [65].

Gas production analyses indicated CO_2_ production, but again no H_2_ was detected. Interestingly, the qCO_2_ values detected at any formate concentration were much lower compared to the growth experiments (compare Table 2 and Table 4). However, this finding is in strong contrast to the growth experiments, as the final CO_2_ concentrations during formate metabolisation were much higher (compare Figure 2 to Figure 4). These observations suggest that the metabolism of *D. amylolyticus* was retarded during the whole cell conversion experiment and that the released CO_2_ could not be assimilated into biomass. The results of the whole cell conversion experiments revealed that the CO_2_, citrate, acetate, and ethanol play key roles in the functioning of the formate metabolism of *D. amylolyticus*.

Based on the obtained metabolite excretion profile, we examined the bioenergetics of formate conversion. The results are shown in Appendix A. ∆G^0′^ and ∆G^0^’/formate at 25 °C and 80 °C for the anaerobic production of formaldehyde + CO_2_, H_2_ + CO_2_, ethanol + CO_2_, acetate + CO_2_ and acetate + ethanol + CO_2_ from formate. ∆G^0’^/formate is given to compare the calculated values based on 1 mol of used formate. Negative values of ∆G^0^’ show that a reaction is thermodynamically favourable under the given conditions. The formation of formaldehyde + CO_2_ from formate is thermodynamically not favourable. Nevertheless, this does not exclude the proposed formaldehyde pathway, as formaldehyde is not an end product. When formaldehyde is converted into other substances after formation, for example for the assimilation into biomass, the complete reaction has to be considered. Ethanol and acetate were found to be the main end products in whole cell conversion with formate as only carbon source. The results show, that these reactions are thermodynamically favourable under the given conditions. The formation of acetate and ethanol out of formate provides small amounts of energy, consistent with the slow growth of *D. amylolyticus*. As the results show, temperature has little influence on ∆G^0^’ for these reactions in the range of 25 to 80 °C.

To understand the formate metabolism of *D. amylolyticus*, we retrieved the amino acid sequences of characterized enzyme complexes, which are known to be involved in different formate utilization pathways: PFL, HDCR, Fdh, and FoDH. We then used the protein sequences of the formate utilization pathway-related enzyme complexes from selected microorganisms (*T. onnurineus*, *P. furious*, *E. coli*, *M. barkeri*, *T. kivui*, and *A. woodii*), to identify their homologous proteins in the genomes of *D. amylolyticus*. Based on these analyses, formate-related pathways were predicted in *D. amylolyticus*. Furthermore, we examined if the genetic arrangement of these sequences resembled the one for *D. amylolyticus*. The results of this analysis are shown in Figure 5.

The protein subunits of the HDCR complex of *A. woodii* are one of the microbial formate assimilation mechanisms. According to our ortho-group analysis, only the electron transfer subunits (Awo_c08200, Awo_c08230, Awo_c08250) of *A. woodii* belong to the same ortho-group as *D. amylolyticus* (Desfe_1134), which indicates that both may have the same function. On the other hand, FdhF1/2 and HydA proteins were located in different ortho-groups and were not identified in the genome of *D. amylolyticus*.

We then hypothesised whether *D. amylolyticus* possesses the genes of PFL and PFL-AE [30]. While, the Desfe_1164 sequence of *D. amylolyticus* showed similarities with *pfl*A of *E. coli* [32], Desfe_0583 resembled TON_0415 of *T. onnurineus* and Awo_c27600 sequence of *A. woodii*, which are annotated as PFL-AE [28]. A comparison of sequences of PFL and PFL-AE proteins from *A. woodii* with *D. amylolyticus* revealed that the alignment is significant concerning E-value and identity (Figure 5, Appendix A). The PFL (or formate C-acetyltransferase) (EC 2.3.1.54) present in *E. coli* and *A. woodii* could not be detected in *D. amylolyticus*. Even though the similar PFL systems were not detected in *D. amylolyticus*, our analysis revealed that the genome harbours a high number of radical SAM proteins (Desfe_0007, Desfe_0149, Desfe_0201, Desfe_0288, Desfe_0298, Desfe_0313, Desfe_0363, Desfe_0369, Desfe_0376, Desfe_0576, Desfe_0583, Desfe_0693, Desfe_0860, Desfe_1164, Desfe_0130, Desfe_0177, Desfe_1197, Desfe_1234) [31]. This might indicate that the PFL function is supported by another radical SAM protein, which is not similar to the PFL of *E. coli* or *A. woodii*. This finding is also not surprising considering that very few archaea possess PFL [82,83]. However, *D. amylolyticus* possesses PFL-AE genes (Desfe_0583, Desfe_1164, and Desfe_1234), and it was recently shown that the PFL-AE homolog in *T. onnurineus* NA1 is strongly upregulated during growth on formate [56].

We then investigated the *D. amylolyticus* genome with respect to the hydrogenase gene clusters of *T. onnurineus* to identify possible orthologous proteins. Our sequence alignment showed that *D. amylolyticus* possesses one multimeric membrane bound hydrogenase subcluster (*mfh*) Desfe_1135-1141), and two H^+^/Na^+^ antiporters (*mnh*) (Desfe_0344-0350 and Desfe_1085-1091) that are similar to subcluster *mfh2* and *mnh1-mnh2* of *T. onnurineus*. Regarding the *fdh* subcluster, which contains *fdh* and electron transfer genes, we were able to identify only the electron transfer gene (Desfe_1134) in *D. amylolyticus.* Additionally, all protein sequences of *fdh* subcluster belonging to molybdopterin Pfam family of proteins were downloaded for all species from Pfam, including the aforementioned strains, and compared them with the *D. amylolyticus* genome. Unfortunately, we couldn’t detect any *fdh* genes in *D. amylolyticus* (Figure 5, Appendix A). However, the auxiliary proteins involved in hydrogenase maturation (Desfe_0501, Desfe_0337, Desfe_0339), which were found to be homologous to hydrogenase maturation proteins of *T. onnurineus* (*Hyc I*; TON_0263, *Hyp F*; TON_0287, *Hyp E*; TON_0286), were identified in the genome of *D. amylolyticus*. Several studies conducted with *E. coli* and *T. onnurineus* resulted in the identification of known auxiliary proteins involved in hydrogenase maturation [28,56,84]. These studies showed that the expression of the *hyc* operon, which contains hydrogenase maturation genes, was upregulated in formate grown cells. This could indicate that the auxiliary proteins of *D. amylolyticus* (Desfe_0501, Desfe_0337, Desfe_0339) might also have an important role in formate metabolism in *D. amylolyticus*.

Furthermore, we examined whether the necessary genes to generate ATP in *T. onnurineus* can be identified in the genome of *D. amylolyticus*. In *T. onnurineus,* a hydrogenase is coupled to an H^+^ antiporter involved in the formation of a Na^+^ gradient, which can be used for ATP generation [25,57]. Despite the fact that *D. amylolyticus* might possess an orthologous membrane bound hydrogenase, which is coupled to a H^+^ antiporter in *T. onnurineus*, the *fdh* subcluster genes could not be detected in the genome. This analysis also supports the experimental observations that *D. amylolyticus* did not produce any H_2_ from formate. However, is must be noted that *D. amylolyticus* produced ppm amounts of H_2_ from cellulose and glucose during batch fermentation in bioreactors [64] and in previously published closed batch cultivations [63]. On the other hand, H_2_ was not detectable during our recent closed batch experiments [65]. Hence, such an ATP synthesis system in *D. amylolyticus* remains to be detected.

The gluconeogenesis and pentose phosphate (PP) pathway enzymes of *T. onnurineus* were already investigated during formate utilisation. It was shown that the glyceraldehyde-3-phosphate dehydrogenase (TON_0639), 2-phosphoglycerate kinase, (TON_0742), fructose bisphosphatase (TON_1497), ribose-5-phosphate isomerase (TON_0168), adenine phosphoribosyl transferase (TON_0120), AMP phosphorylase homolog DeoA (TON_1062), and 3-hexulose-6-phosphate synthase/6-phospho-3-hexuloisomerase (HPS/PHI) (TON_0336) were upregulated during growth on formate in *T. onnurineus* [56]. It was also demonstrated that gluconeogenesis and the PP pathway products, such as ribose-5-phosphate and NADPH were favoured when formate was used as a substrate [56].

The genome of *D. amylolyticus* encodes for several NADH generating genes, however, according to the results of this study, formate oxidation is not coupled to H_2_ evolution and PFL encoding genes are missing in the genome of *D. amylolyticus*.

## 4. Discussion

Based on the above analysis, we hypothesise that the organism might operate the central metabolism with formaldehyde rather than formate. Therefore, we propose that two formate-metabolising reactions might occur in *D. amylolyticus*. First, the reduction of formate with coenzyme A (CoA) to formyl-CoA, which might be catalysed by acetyl-CoA synthetase, and furthermore, the conversion of formaldehyde with the support of an acetylating acetaldehyde dehydrogenase [58]. Second, the direct conversion of formate to formaldehyde through NADH via FoDH [61].

Our analysis showed that *D. amylolyticus* harbours the following proteins: Desfe_0278, Desfe_0067, and Desfe_0019-Desfe_1240 for the enzymes formyl/acetyl transferase (F/AT) [3.1.2.10], (catalyses the reaction: formate <=> formyl-CoA), acylating acetaldehyde dehydrogenase (ADH) [1.2.1.10] (catalyses formyl-CoA <=> formaldehyde), and glutathione-independent formaldehyde dehydrogenase [1.2.1.46] (catalyses formaldehyde + NAD^+^ + H_2_O <=> Formate + NADH + H^+^), respectively (Appendix A). The generated formaldehyde can be assimilated with the oxidative PP pathway (OPPP) which is an efficient route for the assimilation of one-carbon compounds into the central carbon metabolism [85,86,87]. OPPP enzymes catalyse the oxidation of glucose-6-phosphate (G6P) to ribulose-5-phosphate (Ru5P), which was recently shown in halophilic archaea [88]. However, the genes encoding some of the OPPP enzymes, glucose-6-phosphate dehydrogenase and 6-phosphogluconate dehydrogenase are missing in the genome of *D. amylolyticus*. In several Archaea, it has been shown that the conventional PP pathway is incomplete [89]. Moreover, it has been confirmed through biochemical and genome analyses of Archaea that ribulose monophosphate pathway (RuMP) substitutes for the incomplete PP pathway [90]. The generated formaldehyde can be assimilated with inclusion of the synthesis of Ru5P from fructose 6-phosphate (F6P) through the reverse reaction of formaldehyde fixation by HPS/PHI via the RuMP (Figure 6).

The RuMP provided metabolic precursors for the anabolism. The key enzymes are HPS (Desfe_0079), catalysing the reaction from formaldehyde to arabino-3-hexulose-6-phosphate and PHI (Desfe_0297), which catalyses the isomerization of arabino-3-hexulose-6-phosphate to fructose 6-phosphate (F6P). Further, F6P can be metabolised and generate Ru5P by the bifunctional activity of HPS/PHI (Appendix A). The required energy can be substituted by the assimilation of CO_2_ together with ribulose 1,5 bis-phosphate to 3-phosphoglycerate via the activity of RuBisCO [91]. The produced 3-phosphoglycerate could be used for ATP production via glycolysis, while ATP and CO_2_ production can occur via incomplete/pseudo TCA cycle [56]. However, our hypothesis would need to be validated through the combined approach of transcriptomics and proteomics—an endeavour of importance and of high dignity—considering the fastidious growth characteristics of this fascinating organism.

## 5. Conclusions

Through a combined approach of in silico analyses and physiological experiments, we could show that *D. amylolyticus* has the ability to metabolise formate as carbon and energy substrate. *D. amylolyticus* grew at similar µ on formate and glucose, which suggests that this organism faces inherent growth limitations, independent of the supplied carbon and energy substrate concentration. Supported by our experiments and analyses, we propose that the identified homologs of formaldehyde dehydrogenase genes are the only currently-known possibility allowing the metabolisation of formate. Therefore, we would like to raise the possibility that *D. amylolyticus* uses FoDH as a formate assimilation mechanism to produce formaldehyde, and that formaldehyde is subsequently assimilated into biomass through the RuMP. We consequently demonstrate that the CO_2_ released during growth on formate is efficiently assimilated into biomass. Our findings shed new light on the metabolic versatility of the archaeal phylum Crenarchaeota and offers insight into a putative new C1 assimilation pathway in prokaryotes.

## Figures and Tables

**Figure 1 microorganisms-08-00454-f001:**
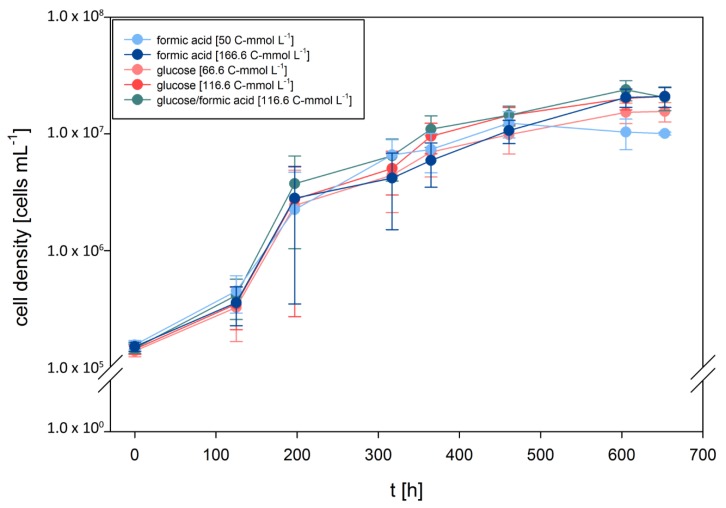
Growth curves of *D. amylolyticus* on formate, glucose, and glucose/formate at different concentrations. A slightly higher µ could be obtained when glucose/formate was used as substrate. All the substrate concentrations are given as C-mmol L^−1^. A negative (un-inoculated) control and positive–negative (inoculated into medium where formic acid was omitted) control were performed in each set and no growth was observed.

**Figure 2 microorganisms-08-00454-f002:**
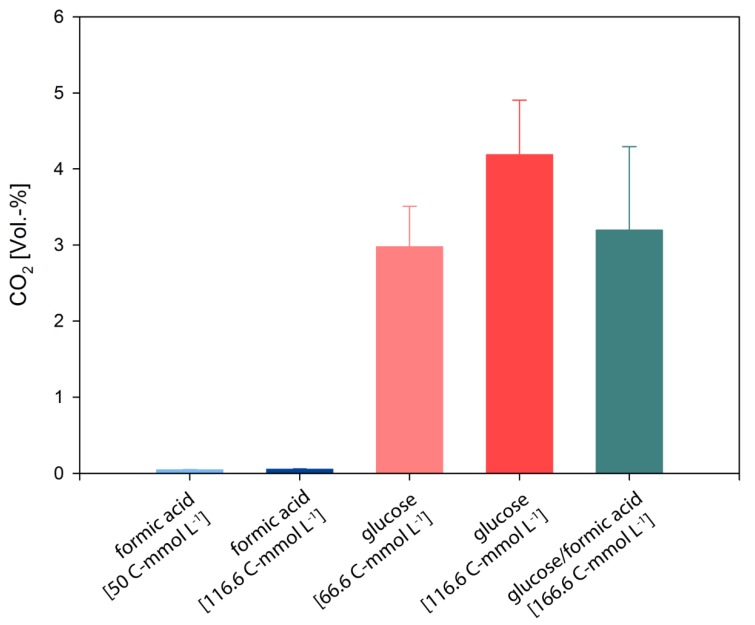
End point gas composition of *D. amylolyticus* grown at different concentrations of formate, glucose, and glucose/formate at the end of the cultivation. The results indicate that the CO_2_ production is very low in the cultures grown on formate compared to cultures grown on glucose or glucose/formate. All the substrate concentrations are given as C-mmol L^−1^.

**Figure 3 microorganisms-08-00454-f003:**
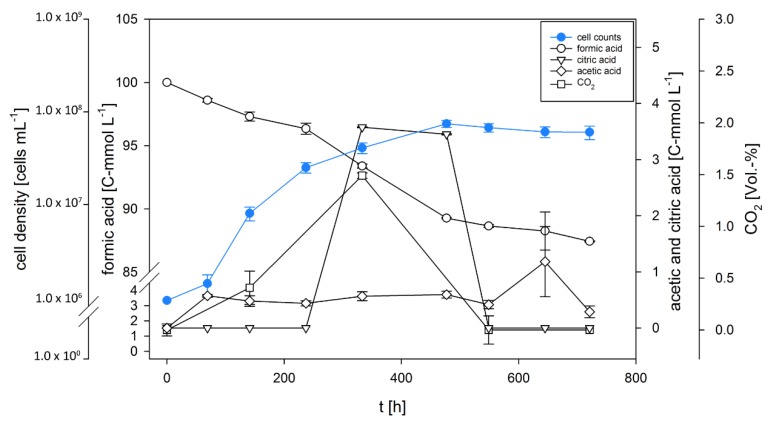
Growth, substrate uptake, and production kinetics of *D. amylolyticus* on 100 C-mmol L^−1^ formate. The results indicate that CO_2_ and citric acid were produced and consumed completely during the cultivation and only after the consumption of CO_2_ and citric acid, acetic acid was produced.

**Figure 4 microorganisms-08-00454-f004:**
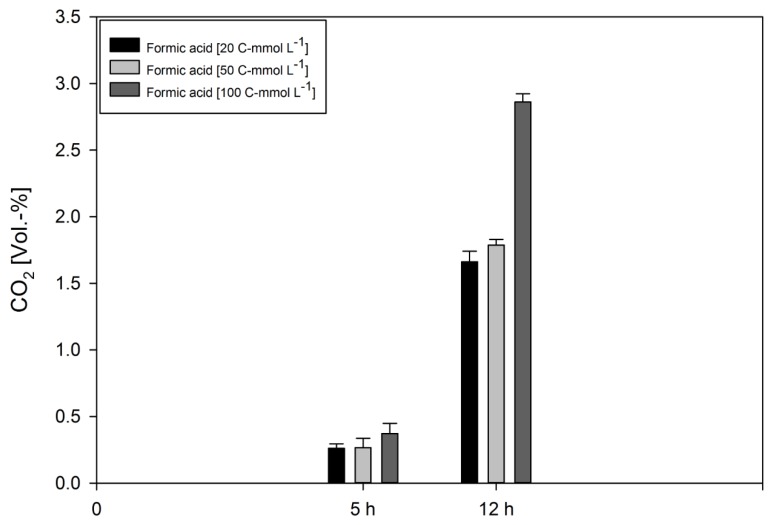
Headspace gas composition of *D. amylolyticus*, from experiments performed in triplicates, at the end points of the whole cell conversion experiment. The experiment was designed and performed to be able to measure the cumulative gas accumulation in the serum bottle headspace. Despite the application of high cell density in the whole cell conversion experiment no H_2_ accumulation was detected.

**Figure 5 microorganisms-08-00454-f005:**
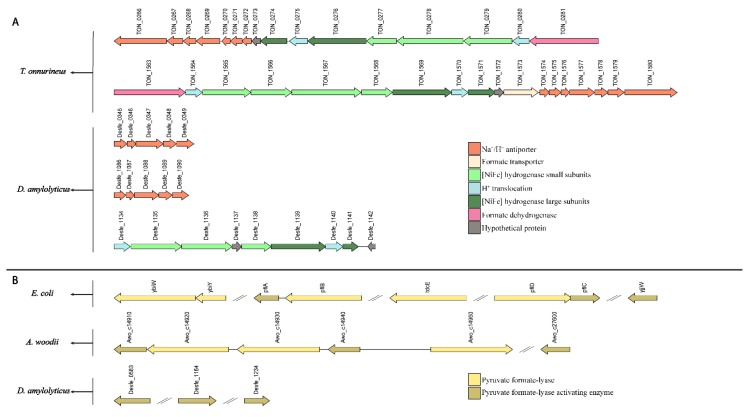
Comparison of genetic organization of (**A**) the fdh complex subunits in *T. onnurineus*, and (**B**) the pyruvate formate lyase (PFL) complex subunits in *E. coli* and *A. woodii* to the genetic organisation in *D. amylolyticus*.

**Figure 6 microorganisms-08-00454-f006:**
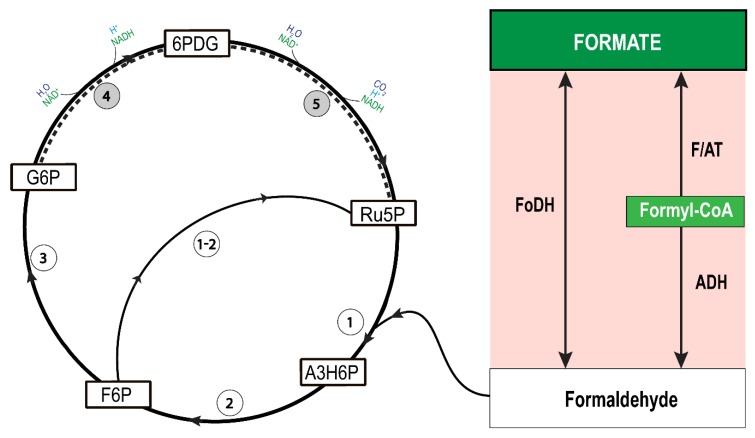
Schematic illustration of the proposed route for formate assimilation in *D. amylolyticus*. The first part of the cycle is formaldehyde production from formate, which might be catalysed by formaldehyde dehydrogenase (FoDH), or formyl/acetyl transferase (F/AT) and aldehyde dehydrogenase (ADH). The second part of the cycle represents formaldehyde assimilation and ribulose 5-phosphate (Ru5P) regeneration via ribulose monophosphate pathway (RuMP) and oxidative pentose phosphate pathway (OPPP). Formaldehyde could be fixed by Ru5P to form D-arabino-3-hexulose-6-phosphate (A3H6P) by 3-hexulose-6-phosphate synthase (HPS) (1) and then isomerized to fructose 6-phosphate (F6P) by 6-phospho-3-hexuloisomerase (PHI) (2). In the genome of *D. amylolyticus,* only gene was found for an HPS-PHI-fused bifunctional enzyme (1-2). F6P is further isomerized to glucose-6-phosphate (G6P) by glucose-6-phosphate isomerase (3). Later, G6P is oxidized to Ru5P by glucose-6-phosphate dehydrogenase (4) and 6-phosphogluconate dehydrogenase (5).

**Table 1 microorganisms-08-00454-t001:** Growth characteristics of *D. amylolyticus.*

Compound and Concentration	μ_max_ [h^−1^]	μ_mean_ [h^−1^]	Final Cell Concentration [cells per mL]
Glucose [66.6 C-mmol L^−1^]	0.033	0.011	1.55·10^7^
Glucose [116.6 C-mmol L^−1^]	0.035	0.012	2.09·10^7^
Formic acid [50 C-mmol L^−1^]	0.032	0.010	1.24·10^7^
Formic acid [116.6 C-mmol L^−1^]	0.032	0.011	2.08·10^7^
Glucose/Formic acid [66 and 50 C-mmol L^−1^]	0.036	0.012	2.38·10^7^

**Table 2 microorganisms-08-00454-t002:** Productivity and yields of *D. amylolyticus* from glucose or formic acid metabolisation during closed batch end-point experiments.

Compound and Concentration	CER [mmol L^−1^ h^−1^] (C-Molar)	qCO_2_ [pmol h^−1^ g^−1^] (C-Molar)	Y_(CO2/s)_ [C-mol/C-mol] ^*^	Y_(Lact/s)_ [C-mol/C-mol] ^*^	Y_(Ac/s)_ [C-mol/C-mol] ^*^	Y_(Form/s)_ [C-mol/C-mol] ^*^	Y_(Glu/s)_ [C-mol/ C-mol] *	Y_(Buty/s)_ [C-mol/C-mol] ^*^	Y_(Citr/s)_ [C-mol/C-mol] ^*^	Y_(Eth/s)_ [C-mol/C-mol] ^*^	Y_(x/s)_ [C-mol/C-mol]	C-Balance ^+^	DoR-Balance ^#^
Glucose [66.6 C-mmol L^−1^]	3.34·10^−5^	2.41·10^−3^	2.64·10^−2^	2.96·10^−1^	5.64·10^−1^	6.14·10^−2^					1.29·10^−4^	94.80%	89.08%
Glucose [116.6 C-mmol L^−1^]	4.04·10^−5^	2.59·10^−3^	4.43·10^−1^	2.48·10^−1^	5.89·10^−1^	8.10·10^−2^					2.42·10^−3^	136.12%	87.82%
Formic acid [50 C-mmol L^−1^]	1.20·10^−6^	1.16·10^−2^	7.20·10^−4^		1.45·10^−1^		8.69·10^−3^	2.15·10^−4^	5.59·10^−2^	5.17·10^−1^	6.34·10^−5^	72.79%	194.37%
Formic acid [116.6 C-mmol L^−1^]	3.48·10^−6^	2.23·10^−4^	5.02·10^−4^		6.21·10^−1^		5.12·10^−2^		1.02·10^−2^	1.59·10^−2^	3.18·10^−5^	69.84%	140.65%

* Y; Yield of product (CO_2_, lactate, acetate, formate, glucose, butyrate, citrate, ethanol, biomass) per substrate (s) consumed. ^+^ Carbon balance. ^#^ Degree of reduction balance.

**Table 3 microorganisms-08-00454-t003:** Productivities and yields of *D. amylolyticus* grown on formic acid during time-point experiments.

Time [h]	CER [mmol L^−1^ h^−1^] (C-Molar)	qCO_2_ [pmol h^−1^ g^−1^] (C-Molar)	Y_(CO2/s)_ [C-mol/C-mol] ^*^	Y_(Ac/s)_ [C-mol/C-mol] ^*^	Y_(Citr/s)_ [C-mol/C-mol] ^*^	Y_(x/s)_ [C-mol/C-mol] ^*^	C-Balance ^+^	DoR-Balance ^#^
69				4.09·10^−1^		1.05·10^−5^	40.93%	81.87%
141	2.12·10^−5^	2.65·10^−6^	2.32·10^−3^	3.73·10^−1^		6.57·10^−5^	37.55%	74.67%
237				4.57·10^−1^		2.75·10^−4^	45.75%	91.53%
333	5.35·10^−5^	1.31·10^−6^	6.04·10^−3^	1.93·10^−1^	1.21·10^0^	1.46·10^−4^	141.12%	220.45%
477				1.44·10^−1^	8.36·10^−1^	1.91·10^−4^	98.05%	154.30%
549	NA ^§^	NA^§^	1.82·10^−3^	6.74·10^−1^		1.15·10^−3^	66.53%	134.93%
645				3.03·10^0^		1.65·10^−3^	302.83%	90.15%
721	NA ^§^	NA^§^	3.53·10^−3^	3.54·10^−1^		7.76·10^−4^	34.11%	70.77%

* Y; Yield of product (CO_2_, acetate, citrate, biomass) per substrate (s) consumed. ^+^ Carbon balance. ^#^ Degree of reduction balance. ^§^ NA; Not available.

**Table 4 microorganisms-08-00454-t004:** Physiological key variables of *D. amylolyticus* obtained from whole cell conversion experiments on formic acid.

Time [h]	Concentration [C-mmol L^−1^]	Formic acid Consumption [%]	CER [mmol L^−1^ h^−1^] (C-Molar)	qCO_2_ [mmol h^−1^ g^−1^] (C-Molar)	Y_(CO2/s)_ ^*^ [C-mol/C-mol]	Y_(Ac/s)_ ^*^ [C-mol/C-mol]	Y_(Buty/s)_ ^*^ [C-mol/C-mol]	Y_(Citr/s)_ ^*^ [C-mol/C-mol]	Y_(Eth/s)_ ^*^ [C-mol/C-mol]
5	100	1.42	2.50·10^−5^	1.60·10^−15^	8.78·10^−5^		3.41·10^−3^		5.72·10^−2^
5	50	0.71	1.79·10^−5^	1.14·10^−15^	2.53·10^−4^		1.97·10^−3^	2.95·10^−2^	
5	20	0.08	1.76·10^−5^	1.13·10^−15^	5.41·10^−3^				
12	100	41.05	1.93·10^−4^	1.23·10^−14^	5.63·10^−5^	3.06·10^−1^		1.43·10^−4^	4.63·10^−4^
12	50	9.35	1.20·10^−4^	7.71·10^−15^	3.09·10^−4^			7.72·10^−3^	2.81·10^−2^
12	20	9.36	1.12·10^−4^	7.17·10^−15^	7.18·10^−5^			1.92·10^−4^	1.26·10^−2^

* Y; Yield of product (CO_2_, acetate, butyrate, citrate, ethanol, biomass) per substrate (s) consumed.

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
