# Peer review of "Formate Utilization by the Crenarchaeon Desulfurococcus amylolyticus"

_microorganisms, 2020, doi:10.3390/microorganisms8030454_

Round 1

Reviewer 1 Report

It is an interesting and well written study. There though some ambiguous points in the design and presentation of the whole cell experiments that should be addressed before acceptance of the study for publication. Specific comments below

Line 133: did you omit yeast extract in all cultures, batch and whole cell when formate was used as the sole carbon source. If yes what was the reason for that?

Line 156-157Q: what was the variation of pressure in the cultures?

Paragr. 2.4: Whole cell conversion experiments. My first assumption was that they are “no growth experiments” but you supplemented the cultures with N. Of course the quantity is minimal compared to the inoculum size and thus taking also into account that the generation time of the microorganism is about 2 h (for μmx0.35h-1) nitrogen would be consumed pretty soon. You should probably elaborate on the design and the purpose of those experiments. What was the pressure of the gaseous phase? Was the initial gaseous phase pure N as in the batch ones? Did you perform cell counts in the experimenst?

Table 1:

  1. Where do you attribute the difference of the μmax value for formate initial concentration of 50mmol/L?
  2. What is the meaning of Yform/Glu, when formate is the sole carbon source?
  3. With formate as the sole carbon source the C balance is incomplete. Can this be attributed to the accumulation of an intracellular metabolite or could there be an extracellular one that was not detected?

Table 3: Please add the units (min) for Time in column 1.

Paragr. 3.2

  1. Lines 326-328, comparison of tables 2 & 4. Citric acid is generally accumulated by different microorgansims under nitrogen limiting conditions. The “whole cell” cultures were supplemented with 0.33g/L NHaCl whereas the batch ones with 0.33g/L NHaCl plus 0.8g/L yeast extract. Taking into account the high initial concentration of ‘whole cell” experiments is would be expected that nitrogen limiting conditions were shorly developed in the cultures.  Couldn’t be some how responsible for the higher citric acid levels in those experiments?
  2. Lines 338-340, comparison of Fig. 2&4: Could this observation be partly attributed to the high pressure (~5atm) maintained in the gasous phase of the batch experiments? What was the pressure of the “whole cell” experiments? If it was 1 atm, the solubility of CO2 in the medium would be 5 time higher in batch experiments. i.e. using Henry’s constant we would expect a concentration of 34mmol/L and 170mmol/L for the batches and the whole cell cultures (assuming 25oC and 1 or 5 atm, respectively. What is the expected productivity of Co2 of formate based on your assumptions and estimations?
  3. Maybe whole cell experiments with glucose should also be conducted in order to enlighten further the metabolism of the microorganism and support the arguments of the study.

Author Response

It is an interesting and well written study. There though some ambiguous points in the design and presentation of the whole cell experiments that should be addressed before acceptance of the study for publication. Specific comments below

Line 133: did you omit yeast extract in all cultures, batch and whole cell when formate was used as the sole carbon source. If yes what was the reason for that?

Reply: Thank you for this comment. Yes, all the experiments were conducted without yeast extract. Initially, it was decided to examine formate utilization as well as the H2 and CO2 productivity only in defined medium (without yeast extract). The reason for decision was that yeast extract is a very expensive source of nutrients and, due to economic reasons, if a renewable energy production processes is envisaged, any extra operational expenditures would need to be carefully balanced during an early bioprocess development stage. However, we discovered that the D. amylolyticus in fact does not produce H2 from formate.

Line 156-157Q: what was the variation of pressure in the cultures?

Reply: Thank you for your question. Pressure range was between 0.2-0.6 bar.

Paragr. 2.4: Whole cell conversion experiments. My first assumption was that they are “no growth experiments” but you supplemented the cultures with N. Of course the quantity is minimal compared to the inoculum size and thus taking also into account that the generation time of the microorganism is about 2 h (for μmx0.35h-1) nitrogen would be consumed pretty soon. You should probably elaborate on the design and the purpose of those experiments. What was the pressure of the gaseous phase? Was the initial gaseous phase pure N as in the batch ones? Did you perform cell counts in the experimenst?

Reply: Thank you for your comment. To keep the cultures anaerobic, N2 was provided. There is not yet any evidence that D. amylolyticus can fix N2 in literature. Besides, GC measurements showed that N2 in fact was not consumed. Pressure range was between 0.2-0.6 bar. And all measurements were optimized according to negative control. The generation time of the microorganism is unfortunately not 2 h. (please refer to Table 1 and Figure 1). Initial gaseous phase was 100% N2. Indeed, we performed cell counts of all of the experiments.

Table 1:
1. Where do you attribute the difference of the μmax value for formate initial concentration of 50mmol/L?

Reply: Thank you for pointing this mistake out! The µmax values are quite homogenous and we corrected the values from 0.022 h-1 to read now 0.032 h-1.

2. What is the meaning of Yform/Glu, when formate is the sole carbon source?

Reply: Thank you for your question. In this manuscript formate and glucose were tested as carbon sources. Yform(glu)/s indicates that, yield of formate produced when glucose (s) was used as carbon source or yield of glucose produced when formate (s) was used as carbon source. We added another column to the table and separated Yform/s and Yglu/s to avoid the confusion.

3. With formate as the sole carbon source the C balance is incomplete. Can this be attributed to the accumulation of an intracellular metabolite or could there be an extracellular one that was not detected?

Reply: Thank you for your question. The question is quite valid and we cannot rule out the possibility of the accumulation of an intracellular metabolite. However we can confidently say that the growth products were measured thoroughly (Table 2) also for the whole cell conversion experiments (Table 4). The tested compounds were citric acid, glucose, formic acid, acetic acid, ethanol, propanol, butyric acid, lactic acid.

Table 3: Please add the units (min) for Time in column 1.

Reply: Thank you for your comment. We added the unit. Time unit is unfortunately hour.

Paragr. 3.2
1. Lines 326-328, comparison of tables 2 & 4. Citric acid is generally accumulated by different microorgansims under nitrogen limiting conditions. The “whole cell” cultures were supplemented with 0.33g/L NHaCl whereas the batch ones with 0.33g/L NHaCl plus 0.8g/L yeast extract. Taking into account the high initial concentration of ‘whole cell” experiments is would be expected that nitrogen limiting conditions were shorly developed in the cultures.  Couldn’t be somehow responsible for the higher citric acid levels in those experiments?

Reply: Thank you for your comment. Firstly, we want to point out that, in our previous study (Reischl et al., 2018, FOLM), microorganism were tested in both with and without yeast extract containing medium. Secondly, D. amylolyticus is an extremely slow growing microorganism, to reach 107 cells per mL, it takes approximately 700 hours. The whole cell conversion (WCC) experiments were only 5 and 12 hours. As it mentioned in the text, WCC experiments were conducted without vitamins and trace elements inside the medium. Even though microorganism was able to grow without addition of vitamins and trace element, which is not possible, 5 and 12 hours is quite limited time for D. amylolyticus to grow.

2. Lines 338-340, comparison of Fig. 2&4: Could this observation be partly attributed to the high pressure (~5atm) maintained in the gasous phase of the batch experiments? What was the pressure of the “whole cell” experiments? If it was 1 atm, the solubility of CO2 in the medium would be 5 time higher in batch experiments. i.e. using Henry’s constant we would expect a concentration of 34mmol/L and 170mmol/L for the batches and the whole cell cultures (assuming 25oC and 1 or 5 atm, respectively. What is the expected productivity of Co2 of formate based on your assumptions and estimations?

Reply: Thank you for your comment. The pressure was always between 0.2-0.6 bar for all experiments. Also the CO2 productivities (CER) were listed in Table 2 and Table 3. We would like to also point out that our experiments were conducted at 80 oC.

3. Maybe whole cell experiments with glucose should also be conducted in order to enlighten further the metabolism of the microorganism and support the arguments of the study.

Reply: Thank you for your comment. This manuscript was intended to investigate the formate metabolism of D. amylolyticus and it shows the formate is utilized and we tried to investigate how formate might be metabolized. Glucose was investigated in another study (not with whole cell conversion experiments, however with metabolic construction and growth experiments) (Reischl et al., 2018, FOLM). We used glucose in this study to compare growth kinetics and the metabolites of formate to see the differences between glucose and formate as carbon source. We think that whole cell conversion experiments for glucose is not the scope of this study. We hope that you agree to our reasoning.

Reviewer 2 Report

The manuscript of Ergal et al. is a valuable contribution to the metabolism of an archaeon. The structure, the English are both good. There are however some questions, and remarks which have to be addressed.
Let us see the remarks in order of their appearance.
- Introduction. It is strange that the expression methylotrophy is not mentioned. The investigated organism is a facultative methylotroph. It is also interesting that the presence of RUBISCO is similarly not mentioned. Even those twoo well known autotrophic carbon-dioxide fixation pathways (the RuMP and the serine pathway) are not discussed, which utilize formaldehyde.
- lines 57-58. ?
- line 83. mol-1!
- lines 119-120. Rephrase!
- lines 126-129. v/v %!
- lines 131-132. Give the correct name of DSMZ!
- lines 137-161. The composition of the growth medium must be described, since this is a key concerning the evaluation of the results. E.g. did the growth medium contain bicarbonates, or carbonates? This is not only pH sensitive and may have influenced the concentration of the detected carbon dioxide, but at the same time serves as carbon source for an autotroph.
- line 158. Rephrase!
- line 191. Rephrase!
- lines 230-236. What was the logic of this choice? Why there were the formerly mentioned methylotrophs similarly not investigated. Please discuss this topic in the Discussion part!
- Table 2. What is CER?
- Figure 3. What about the medium? Where is ethanol?
- lines 336-344. Was carbon dioxide utilized, or bicarbonate, or carbonate anions?
- line 360. But the figure shows only carbon dioxide concentration, and not the composition of the headspace!
- lines 391-396. What is SAM? Is this S-adenosyl-methionine?
- lines 399-416. What about Fe-S proteins?
- line 469. HPS-PHI?
- line 473. Why only RuMP is an anabolic precursor?
- line 478. Is it not ribulose 1, 5 bis-phosphate? Why only here it is mentioned, that this organism harbours RuBisCO?

Author Response

The manuscript of Ergal et al. is a valuable contribution to the metabolism of an archaeon. The structure, the English are both good. There are however some questions, and remarks which have to be addressed.

Let us see the remarks in order of their appearance.

- Introduction. It is strange that the expression methylotrophic is not mentioned. The investigated organism is a facultative methylotroph. It is also interesting that the presence of RUBISCO is similarly not mentioned. Even those twoo well known autotrophic carbon-dioxide fixation pathways (the RuMP and the serine pathway) are not discussed, which utilize formaldehyde.

Reply: Thank you for your comment. D. amylolyticus was initially characterized as an obligately anaerobic, hyperthermophilic, organoheterotrophic archaeon (Perevalova et al., 2005, Susanti et al., 2012). To our knowledge it is not a facultative methylotroph.

Please see the discussion of RuMP and RUBISCO under the “discussion” section.

- lines 57-58. ?

Reply: Thank you for the comment. We cannot see that there would be a change required.

- line 83. mol-1!

Reply: Thank you for your comment. We corrected this mistake.

- lines 119-120. Rephrase!

Reply: Thank you for your comment. The sentence was changed to “Then, mass balance analysis of substrate uptake and product formation kinetics was performed in whole cell conversion experiments.”

- lines 126-129. v/v %!

Reply: Thank you for the suggestion. Actually, in all papers of the Rittmann Group, the concentrations of gases were specified as they were provided in this manuscript. The concentration given in v/v would be 20/80, which results in 25% not 20%. It should read 20/100 which is strictly speaking also not correct, as we would need to write 20 CO2 /(20 CO2+80 N2), which would give 20%. Therefore, we agreed with editors of an engineering journal to write 20 Vol.-% in N2. We therefore kindly ask you to accept our established style of presenting concentrations.

- lines 131-132. Give the correct name of DSMZ!

Reply: Thank you for your comment. We corrected this mistake.

- lines 137-161. The composition of the growth medium must be described, since this is a key concerning the evaluation of the results. E.g. did the growth medium contain bicarbonates, or carbonates? This is not only pH sensitive and may have influenced the concentration of the detected carbon dioxide, but at the same time serves as carbon source for an autotroph.

Reply: Thank you for your comment. We added the medium composition under “2.2 Microorganism and medium composition” section as “Medium contained (per L): 0.33 g of NH4Cl; 0.33 g of KH2PO4; 0.33 g of KCl; 0.44 g of CaCl2·2H2O; 0.70 g of MgCl2·6H2O; 0.50 g of NaCl; 0.80 g of NaHCO3; 0.50 g of Na2S·9H2O; 1 mL of trace elements SL-10; and 10 mL of vitamin solution as previously described [64,65]”. The experiments conducted without formic acid (but rest of the medium kept the same, meaning contained sodium bicarbonate) showed that the organism is utilizing the formate and cannot grow or produce CO2 in the absence of formic acid. (Please see supplementary Figure 1).

- line 158. Rephrase!

Reply: Thank you for your comment. We changed the sentence as “Pressure measurements of the serum bottle headspace were performed with a digital manometer (LEO1-Ei, −1-3bar rel, Keller, Germany) and the measurements were perfromed before samples were taken for microscope analysis.”

- line 191. Rephrase!

Reply: Thank you for your comment. The sentence was changed as “The microplate (Crystal Clear, Greiner Bio One) assay was applied according to manufacturer's instructions a 255 μL reaction volume.”

- lines 230-236. What was the logic of this choice? Why there were the formerly mentioned methylotrophs similarly not investigated. Please discuss this topic in the Discussion part!

Reply: Thank you for your question. We investigated formate utilization pathway- and its related enzyme complexes from selected microorganisms (T. onnurineus, P. furious, E. coli, M. barkeri, T. kivui, and A. woodii). These organisms can utilize formate. Since D. amylolyticus is not described as methylotrophs in the literature, we haven’t investigated other methylotrophs. Incomplete CO2-fixation pathways and comparison with other Methylotrophs were obtained in our previous paper (Reischl et. al., 2018, FOLM).

- Table 2. What is CER?

Reply: Thank you for your question. It was explained in the line 204. “The CO2 evolution rate (CER [mmol L−1 h−1] (C-molar)), the cell specific CO2 productivity (qCO2, cell [mmol cell−1 h−1] (C-molar)) [67] was calculated from the end point gas composition of the non-manipulated serum bottles.”

- Figure 3. What about the medium? Where is ethanol?

Reply: Thank you for your question. Interestingly in the time-point experiments no ethanol was detected. During whole cell conversion experiments ethanol was detected, and very low amounts of ethanol were measured in end-point experiments. The reason for this difference is not clear to us.

- lines 336-344. Was carbon dioxide utilized, or bicarbonate, or carbonate anions?

Reply: Thank you for this question. The carbonate equilibrium is complex and the consumption of a specific species cannot be easily answered since all these species are in equilibrium. Patch-clamping of putative carbonate or bicarbonate transporters or growth experiment under specific pH could be performed.

- line 360. But the figure shows only carbon dioxide concentration, and not the composition of the headspace!

Reply: Thank you for your comment. As it can be seen from the graph, CO2 was given as percentage. The rest was N2. Other gases (H2, CO2, N2, CH4, CO) were measured as it was given in the materials and methods section. None of them was detected but N2 in the headspace.

- lines 391-396. What is SAM? Is this S-adenosyl-methionine?

Reply: Yes. We added the abbreviation in line 65.

- lines 399-416. What about Fe-S proteins?

Reply: We investigated Fe-S proteins in Reischl et al. 2018 Int. J. Hydrogen Energy and did not detect any during our genomic analysis. However, this doesn’t rule out the possibility that these proteins will be detected.

- line 469. HPS-PHI?

Reply: We added 3-hexulose-6-phosphate synthase/6-phospho-3-hexuloisomerase (HPS/PHI) in line 436-437.

- line 473. Why only RuMP is an anabolic precursor?

Reply: Thank you for your question. RuMP is not the only anabolic precursor we propose. For this organism concerning formate utilization and formate metabolism we suggest that the RuMP could possibly provide metabolic precursors for the anabolism.

- line 478. Is it not ribulose 1, 5 bis-phosphate? Why only here it is mentioned, that this organism harbours RuBisCO?

Reply: Thank you very much for pointing this mistake out. This typo was corrected to “ribulose 1, 5 bis-phosphate”.

Round 2

Reviewer 1 Report

Most of my questions were answered adequately. Doubling time is indeed much higher than 2h (I mistakenly made the calculation assuming, μmax=0.35h-1!)

The question about N source consumption in the whole cell experiments was not referring to the N in the atmosphere but in the medium (ammonium salt). but there in no point for it indeed for experiments with duration 5-12 h!

I 'm still though confused about the pressure in the vials

line 145 - Cultures of D. amylolyticus were grown anaerobically at 5·105 Pa1Pa = 10-5 bar. , 5·105 Pa=5 bar!, How can the pressure range be 0.2-0.6 bar?

Moreover with respect to table 3

"We added another column to the table and separated Yform/s and Yglu/s to avoid the confusion.”

Y(Glu/s) [C-mol/C-mol]:How could possibly glucose be produced by formate uptake??

Author Response

Most of my questions were answered adequately. Doubling time is indeed much higher than 2h (I mistakenly made the calculation assuming, μmax=0.35h-1!)

Reply: Thank you for performing the review!

The question about N source consumption in the whole cell experiments was not referring to the N in the atmosphere but in the medium (ammonium salt). But there in no point for it indeed for experiments with duration 5-12 h!

Reply: Thank you!

I 'm still though confused about the pressure in the vials

line 145 - Cultures of D. amylolyticus were grown anaerobically at 5·105 Pa1Pa = 10-5 bar. , 5·105 Pa=5 bar!, How can the pressure range be 0.2-0.6 bar?

Reply: Thank you for pointing out the mistake again. We corrected that. It is reading now 0.2-0.3·105 Pa.

Moreover with respect to table 3

"We added another column to the table and separated Yform/s and Yglu/s to avoid the confusion.”

Y(Glu/s) [C-mol/C-mol]:How could possibly glucose be produced by formate uptake??

Reply: Thank you for the comment. Many organisms growing on C1-compounds must synthesize, e.g. sugars for several metabolic pathways via gluconeogenesis. However, we do not know as to why the organism releases minor amounts of glucose to the medium. For the sake of completeness we had to add this finding to this table.

Reviewer 2 Report

Dear Authors! The uploaded new version of the manuscript lacks at least two figures. The corrections described in the cover letter are not, or partly made (e.g. the name of DSMZ is still not adequate).

If the genome of the investigated organism contains the RubisCO encoding gene, and it is active, then the organism is capable possibly for carbon dioxide fixation. This cannot be neglected, and must be discussed. Moreover its formaldehyde utilization resembles the organism to methylotrophs. I do not think, that quoting simply that it has been described as an organoheterotrophic archaeon, thus it is that, and based on this assumption not investigating the carbon dioxide assimilation, moreover other methylotrophic type traits in not a scientific answer. Do you know other such organisms, which are organoheterotrophs, but do have an active RubisCO, and fix carbon dioxide? 

Author Response

Dear Authors! The uploaded new version of the manuscript lacks at least two figures. The corrections described in the cover letter are not, or partly made (e.g. the name of DSMZ is still not adequate).

Reply: Thank you for pointing out the mistakes/miscommunications during submission process. We supplied the new figures separately during submission and informed the editor, since uploading new figures was not optional (only the manuscript). However, you clearly haven’t received them. Now we added the new figures into the manuscript ourselves. We checked again DSMZ name as well.

If the genome of the investigated organism contains the RubisCO encoding gene, and it is active, then the organism is capable possibly for carbon dioxide fixation. This cannot be neglected, and must be discussed.

Reply: Thank you for your comment. The organism contains a RubisCo encoding gene. However, we don’t know whether it is active or not. In many archaea the Rubisco is involved in the AMP salvage pathway, not in CO2 fixation. Only for Methanospirillum hungatei a possible role in CO2 fixation was shown. The experimental data we provided in our manuscript indicate that produced CO2 was consumed. Furthermore, in our previous study (Reischl et. al., 2018, FOLM), we performed CO2 (as sole carbon) experiments. CO2 was given to the atmosphere and the bottles did not contain any additional carbon source. In this case we couldn’t observe any growth (because an energy substrate was missing). We are afraid that adding a discussion about an incomplete CO2-fixation pathway in detail, might be redundant and partially plagiating to our previous paper. However, as you pointed out, it had to be mentioned, and considered for the metabolism of this fascinating microorganism. This was the reason we mentioned RubisCO in the manuscript. Moreover, we want to point out that the manuscript is not about CO2 fixation but about formate utilization.

Moreover its formaldehyde utilization resembles the organism to methylotrophs. I do not think, that quoting simply that it has been described as an organoheterotrophic archaeon, thus it is that, and based on this assumption not investigating the carbon dioxide assimilation, moreover other methylotrophic type traits in not a scientific answer. Do you know other such organisms, which are organoheterotrophs, but do have an active RubisCO, and fix carbon dioxide?

Reply: Thank you for this comment. Please let us formulate our explanation again. Your question is a valid question that we asked ourselves too while studying this microorganism. It would have been oblivious not to investigate the C1 fixation possibilities. Maybe we lacked expressing our findings, since the findings allowed us to eliminate this question in a very early stage of this study.

Formaldehyde utilization might resemble this organism to methylotrophs. However, when we look at the enzymes involved in the metabolism of methylotrophs, this organism lacks the encoding genes of the pathway. E.g. genes encoding methyltransferase, methanol dehydrogenase and even formate dehydrogenase are not present in the genome of D. amylolyticus.

T. onnurineus NA1 retains the metabolic pathways for organotrophic growth as D. amylolyticus. T. onnurineus NA1 also has metabolic pathways for carboxydotrophic growth, an active RubisCO and it can fix CO2. This was one of the main reasons, we have chosen T. onnurineus as one of the organism to compare metabolic traits. In case of T. onnurineus, there is no proof that T. onnurineus has PFL, the same accounts for D. amylolyticus. However, energy can be gained from breakdown of formate to H2 + CO2 in case of T. onnurineus and energy conservation occurs through a proton motive gradient. This is the most fascinating with regard to the physiology of D. amylolyticus, since we do not detect any H2 production, the enigma of energy conservation in this organism hass to be solved. Without any proof of complete PFL, complete FDH and complete CO2 fixation; how D. amylolyticus can fix C1 and gain energy must unfortunately remain unanswered at the moment.

Round 3

Reviewer 2 Report

Dear Authors!

The name of DSMZ in German is: Deutsche Sammlung von Mikroorganismen und Zellkulturen GmbH...

Author Response

Response to reviewer 2:

The name of DSMZ in German is: Deutsche Sammlung von Mikroorganismen und Zellkulturen GmbH

Reply: Thank you for this comment.

Lines 131-133: „Deutsche Stammsammlung von Mikroorganismen und Zellkulturen“ was changed to „Deutsche Sammlung von Mikroorganismen und Zellkulturen GmbH“